# Trajeglish: Traffic Modeling as Next-Token Prediction

**Jonah Philion**[1,2,3]**, Xue Bin Peng**[1,4]**, Sanja Fidler**[1,2,3]
[1]NVIDIA, [2]University of Toronto, [3]Vector Institute, [4]Simon Fraser University
{jphilion, japeng, sfidler}@nvidia.com

## Abstract

A longstanding challenge for self-driving development is simulating dynamic driving scenarios seeded from recorded driving logs. In pursuit of this functionality, we apply tools from discrete sequence modeling to model how vehicles, pedestrians and cyclists interact in driving scenarios. Using a simple data-driven tokenization scheme, we discretize trajectories to centimeter-level resolution using a small vocabulary. We then model the multi-agent sequence of discrete motion tokens with a GPT-like encoder-decoder that is autoregressive in time and takes into account intra-timestep interaction between agents. Scenarios sampled from our model exhibit state-of-the-art realism; our model tops the Waymo Sim Agents Benchmark, surpassing prior work along the realism meta metric by 3.3% and along the interaction metric by 9.9%. We ablate our modeling choices in full autonomy and partial autonomy settings, and show that the representations learned by our model can quickly be adapted to improve performance on nuScenes. We additionally evaluate the scalability of our model with respect to parameter count and dataset size, and use density estimates from our model to quantify the saliency of context length and intra-timestep interaction for the traffic modeling task.

## 1 Introduction

In the short term, self-driving vehicles will be deployed on roadways that are largely populated by human drivers. For these early self-driving vehicles to share the road safely, it is imperative that they become fluent in the ways people interpret and respond to motion. A failure on the part of a self-driving vehicle to predict the intentions of people can lead to overconfident or overly cautious planning. A failure on the part of a self-driving vehicle to communicate to people its own intentions can endanger other road users by surprising them with uncommon maneuvers.

In this work, we propose an autoregressive model of the motion of road users that can be used to simulate how humans might react if a self-driving system were to choose a given sequence of actions. At test time, as visualized in Fig. 1, the model functions as a policy, outputting a categorical distribution over the set of possible states an agent might move to at each timestep. Iteratively sampling actions from the model results in diverse, scene-consistent multi-agent rollouts of arbitrary length. We call our approach Trajeglish ("tra-JEG-lish") due to the fact that we model multi-agent *traje*ctories as a sequence of discrete tokens, similar to the representation used in language modeling, and to make an analogy between how road users use vehicle motion to communicate and how people use verbal languages, like En*glish*, to communicate.

A selection of samples from our model is visualized in Fig. 2. When generating these samples, the model is prompted with only the initial position and heading of the agents, in contrast to prior work that generally requires at least one second of historical motion to begin sampling. Our model generates diverse outcomes for each scenario, while maintaining the scene-consistency of the trajectories. We encourage readers to consult our project page for videos of scenarios sampled from our model in full control and partial control settings, as well as longer rollouts of length 20 seconds.

Our main contributions are:

- A simple data-driven method for tokenizing trajectory data we call "k-disks" that enables us to tokenize the Waymo Open Dataset (WOMD) (Ettinger et al., 2021) at an expected discretization error of 1 cm using a small vocabulary size of 384.

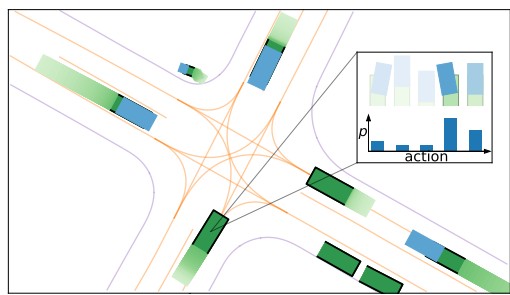

Figure 1: **Inputs and outputs** At a given timestep, our model predicts a distribution over a fixed set of $V$ states defined relative to an agent's current location and heading, and conditions on map information, actions from all previous timesteps (green), and any actions that have already been chosen by other agents within the current timestep (blue). We model motion of all agents relevant to driving scenarios, including vehicles, pedestrians, and cyclists.

- A transformer-based architecture for modeling sequences of motion tokens that conditions on map information and one or more initial states per agent. Our model outputs a distribution over actions for agents one at a time which we show is ideal for interactive applications.
- State-of-the-art quantitative and qualitative results when sampling rollouts given real-world initializations both when the traffic model controls all agents in the scene as well as when the model must interact with agents outside its control.

We additionally evaluate the scalability of our model with respect to parameter count and dataset size, visualize the representations learned by our model, and use density estimates from our model to quantify the extent to which intra-timestep dependence exists between agents, as well as to measure the relative importance of long context lengths for traffic modeling (see Sec. 4.3).

## 1.1 RELATED WORK

Our work builds heavily on recent work in imitative traffic modeling. The full family of generative models have been applied to this problem, including VAEs (Suo et al., 2021; Rempe et al., 2021), GANs (Igl et al., 2022), and diffusion models (Zhong et al., 2022; Jiang et al., 2023). While these approaches primarily focus on modeling the multi-agent joint distribution over future trajectories, our focus in this work is additionally on building reactivity into the generative model, for which the factorization provided by autoregression is well-suited. For the structure of our encoder-decoder, we draw inspiration from Scene Transformer (Ngiam et al., 2021) which also uses a global coordinate frame to encode multi-agent interaction, but does not tokenize data and instead trains their model with a masked regression strategy. A limitation of regression is that it's unclear if the Gaussian or Laplace mixture distribution is flexible enough to represent the distribution over the next state, whereas with tokenization, we know that all scenarios in WOMD are within the scope of our model, the only challenge is learning the correct logits. A comparison can also be made to the behavior cloning baselines used in Symphony (Igl et al., 2022) and "Imitation Is Not Enough" (Lu et al., 2023) which also predict a categorical distribution over future states, except our models are trained directly on pre-tokenized trajectories as input, and through the use of the transformer decoder, each embedding receives supervision for predicting the next token as well as all future tokens for all agents in the scene. In terms of tackling the problem of modeling complicated continuous distributions by tokenizing and applying autoregression, our work is most similar to Trajectory Transformer (Janner et al., 2021) which applies a fixed-grid tokenization strategy to model state-action sequences for RL. Finally, our work parallels MotionLM (Seff et al., 2023) which is concurrent work that also uses discrete sequence modeling for motion prediction, but targets 1- and 2-agent online interaction prediction inistead of $N$-agent offline closed-loop simulation.

## 2 IMITATIVE TRAFFIC MODELING

In this section, we show that the requirement that traffic models must interact with all agents at each timestep of simulation, independent of the method used to control each of the agents, imposes certain structural constraints on how the multi-agent future trajectory distribution is factored by imitative traffic models. Similar motivation is provided to justify the conditions for submissions to the WOMD sim agents benchmark to be considered valid closed-loop policies (Montali et al., 2023).

We are given an initial scene with $N$ agents, where a *scene* consists of map information, the dimensions and object class for each of the $N$ agents, and the location and heading for each of the

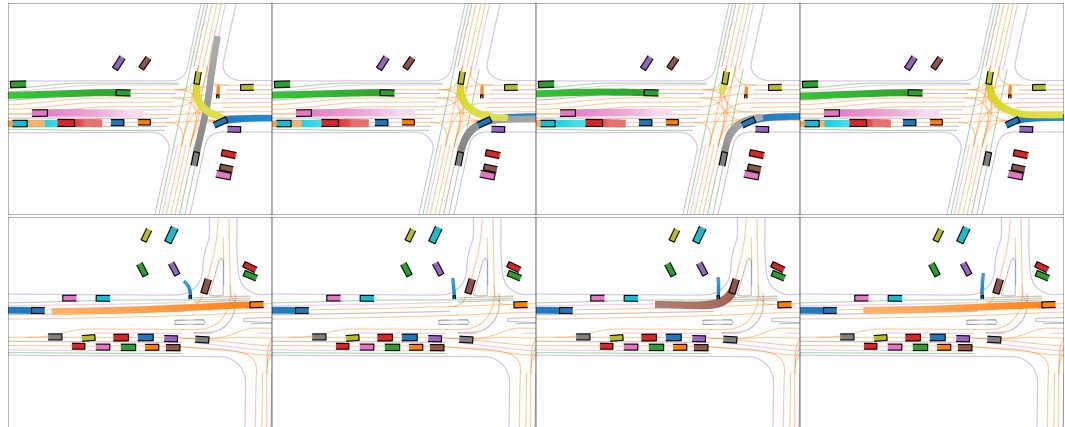

Figure 2: **Trajeglish** Visualizations of samples from our model. Rollouts within each row are given the same single-timestep initialization, outlined in black. Future trajectories become lighter for timesteps farther into the future. Note that while some tracks overlap in the figure, they do not overlap when time is taken into account; there are no collisions in these rollouts. Videos are available on our project page.

agents for some number of timesteps in the past. For convenience, we denote information about the scene provided at initialization by $c$. We denote the *state* of a vehicle $i$ at future timestep $t$ by $s_t^i \equiv (x_t^i, y_t^i, h_t^i)$ where $(x, y)$ is the center of the agent's bounding box and $h$ is the heading. For a scenario of length $T$ timesteps, the distribution of interest for traffic modeling is given by

$$p(s_1^1, ..., s_1^N, s_2^1, ..., s_2^N, ..., s_T^1, ..., s_T^N \mid c). \tag{1}$$

We refer to samples from this distribution as *rollouts*. In traffic modeling, our goal is to sample rollouts under the restriction that at each timestep, a black-box autonomous vehicle (AV) system chooses a state for a subset of the agents. We refer to the agents controlled by the traffic model as "non-player characters" or NPCs. This interaction model imposes the following factorization of the joint likelihood expressed in Eq. 1

$$
\begin{aligned}
&p(s_1^1, ..., s_1^N, s_2^1, ..., s_2^N, ..., s_T^1, ..., s_T^N \mid c) \\
&= \prod_{1 \le t \le T} p(s_t^{1...N_0} | c, s_{1...t-1}) \underbrace{p(s_t^{N_0+1...N} \mid c, s_{1...t-1}, s_t^{1...N_0})}_{\text{NPCs}}
\end{aligned}
\tag{2}
$$

where $s_{1...t-1} \equiv \{s_1^1, s_1^2, ..., s_{t-1}^N\}$ is the set of all states for all agents prior to timestep $t$, $s_t^{1...N_0} \equiv \{s_t^1, ..., s_t^N\}$ is the set of states for agents 1 through $N$ at time $t$, and we arbitrarily assigned the agents out of the traffic model's control to have indices $1, ..., N_0$. The factorization in Eq. 2 shows that we seek a model from which we can sample an agent's next state conditional on all states sampled in previous timesteps as well as any states already sampled at the current timestep.

We note that, although the real-world system that generated the driving data involves independent actors, it may still be important to model the influence of actions chosen by other agents at the same timestep, a point we expand on in Appendix A.1. While intra-timestep interaction between agents is weak in general, explicitly modeling this interaction provides a window into understanding cases when it is important to consider for the purposes of traffic modeling.

## 3   METHOD

In this section, we introduce Trajeglish, an autoregressive generative model of dynamic driving scenarios. Trajeglish consists of two components. The first component is a strategy for discretizing, or "tokenizing" driving scenarios such that we model exactly the conditional distributions required by the factorization of the joint likelihood in Eq. 2. The second component is an autoregressive transformer-based architecture for modeling the distribution of tokenized scenarios.

Important features of Trajeglish include that it preserves the dynamic factorization of the full likelihood for dynamic test-time interaction, it accounts for intra-timestep coupling across agents, and

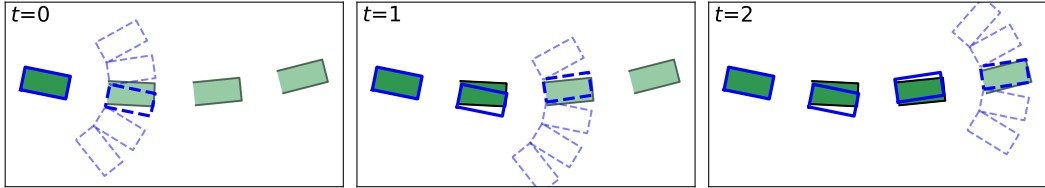

Figure 3: **Tokenization** We iteratively find the token with minimum corner distance to the next state. An example trajectory is shown in green. The raw representation of the tokenized trajectory is shown as boxes with blue outlines. States that have yet to be tokenized are light green. Token templates are optimized to minimize the error between the tokenized trajectories and the raw trajectories.

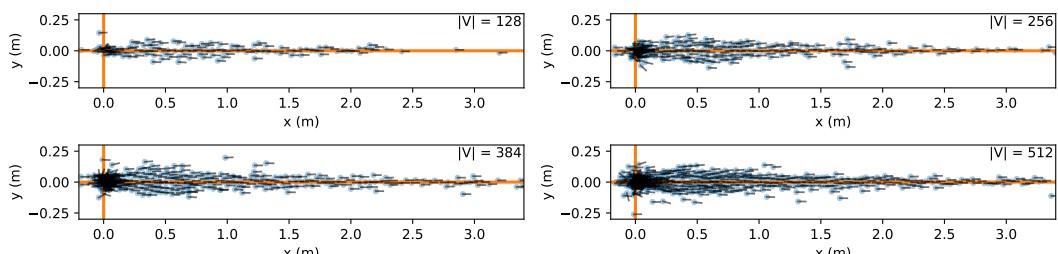

Figure 4: **Raw motion token representation** We plot the raw representation of action sets extracted with k-disks for $|V| \in \{128, 256, 384, 512\}$. Agents sample one of these actions at each timestep.

it enables both efficient sampling of scenarios as well as density estimates. While sampling is the primary objective for traffic modeling, we show in Sec. 4.3 that the density estimates from Trajeglish are useful for understanding the importance of longer context lengths and intra-timestep dependence. We introduce our tokenization strategy in Sec. 3.1 and our autoregressive model in Sec. 3.2.

## 3.1 TOKENIZATION

The goal of tokenization is to model the support of a continuous distribution as a set of $|V|$ discrete options. Given $\boldsymbol{x} \in \mathbb{R}^n \sim p(\boldsymbol{x})$, a *tokenizer* is a function that maps samples from the continuous distribution to one of the discrete options $f : \mathbb{R}^n \to V$. A *renderer* is a function that maps the discrete options back to raw input $r : V \to \mathbb{R}^n$. A high-quality tokenizer-renderer pair is one such that $r(f(\boldsymbol{x})) \approx \boldsymbol{x}$. The continuous distributions that we seek to tokenize for the case of traffic modeling are given by Eq. 1. We note that these distributions are over single-agent states consisting of only a position and heading. Given the low dimensionality of the input data, we propose a simple approach for tokenizing trajectories based on a fixed set of state-to-state transitions.

**Method** Let $\boldsymbol{s}_0$ be the state of an agent with length $l$ and width $w$ at the current timestep. Let $\boldsymbol{s}$ be the state at the next timestep that we seek to tokenize. We define $V = \{\boldsymbol{s}_i\}$ to be a set of *template actions*, each of which represents a change in position and heading in the coordinate frame of the most recent state. We use the notation $a_i \in \mathbb{N}$ to indicate the index representation of token template

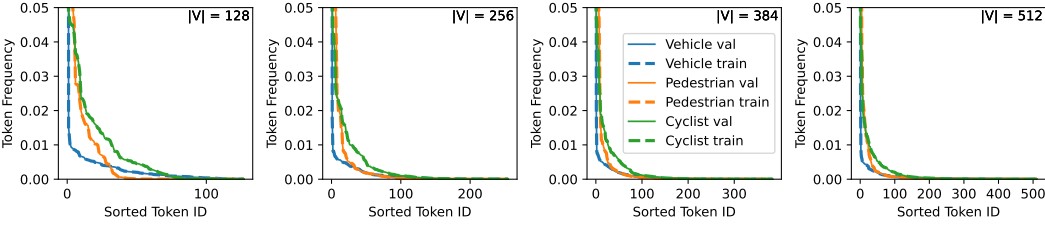

Figure 5: **Token frequency** We plot the frequency that each token appears in the validation and training sets. Note that we sort the tokens by their frequency for each class individually for the ID. Increasing the vocabulary size increases the resolution but also results in a longer tail. The distribution of actions on the training set and validation set match closely.

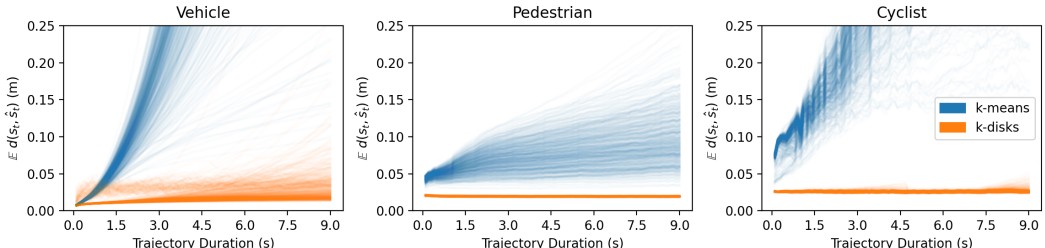

Figure 6: **K-means vs. k-disks** We plot the average discretization error for multiple template sets sampled from k-means and k-disks with $|V| = 384$. Alg. 1 consistently samples better template sets than k-means.

$s_i$ and $\hat{s}$ to represent the raw representation of the tokenized state $s$. Our tokenizer $f$ and renderer $r$ are defined by

$$f(s_0, s) = a_{i^*} = \arg\min_i d_{l,w}(s_i, \text{local}(s_0, s)) \tag{3}$$

$$r(s_0, a_i) = \hat{s} = \text{global}(s_0, s_i) \tag{4}$$

where $d_{l,w}(s_0, s_1)$ is the average of the L2 distances between the ordered corners of the bounding boxes defined by $s_0$ and $s_1$, "local" converts $s$ to the local frame of $s_0$, and "global" converts $s_{i^*}$ to the global frame out of the local frame of $s_0$. We use $d_{l,w}(\cdot, \cdot)$ throughout the rest of the paper to refer to this mean corner distance metric. Importantly, in order to tokenize a full trajectory, this process of converting states $s$ to their tokenized counterpart $\hat{s}$ is done iteratively along the trajectory, using tokenized states as the base state $s_0$ in the next tokenization step. We visualize the procedure for tokenizing a trajectory in Fig. 3. Tokens generated with our approach have three convenient properties for the purposes of traffic modeling: they are invariant across coordinate frames, invariant under temporal shift, and they supply efficient access to a measure of similarity between tokens, namely the distance between the raw representations. We discuss how to exploit the third property for data augmentation in Sec. A.2.

**Optimizing template sets** We propose an easily parallelizable approach for finding template sets with low discretization error. We collect a large number of state transitions observed in data, sample one of them, filter transitions that are within $\epsilon$ meters, and repeat $|V|$ times. Pseudocode for this algorithm is included in Alg. 1. We call this method for sampling candidate templates "k-disks" given its similarity to k-means++, the standard algorithm for seeding the anchors k-means (Arthur & Vassilvitskii, 2007), as well as the Poisson disk sampling algorithm (Cook, 1986). We visualize the template sets found using k-disks with minimum discretization error in Fig. 4. We verify in Fig. 5 that the tokenized action distribution is similar on WOMD train and validation despite the fact that the templates are optimized on the training set. We show in Fig. 6 that the discretization error induced by templates sampled with k-disks is in general much better than that of k-means, across agent types. A comprehensive evaluation of k-disks in comparison to baselines is in Sec. A.3.

### 3.2 MODELING

The second component of our method is an architecture for learning a distribution over the sequences of tokens output by the first. Our model follows an encoder-decoder structure very similar to those used for LLMs (Vaswani et al., 2017; Radford et al., 2019; Raffel et al., 2019). A diagram of the model is shown in Fig. 7. Two important properties of our encoder are that it is not equivariant to choice of global coordinate frame and it is not permutation equivariant to agent order. For the first property, randomizing the choice of coordinate frame during training is straightforward, and sharing a global coordinate frame enables shared processing and representation learning across agents. For the second property, permutation equivariance is not actually desirable in our case since the agent order encodes the order in which agents select actions within a timestep; the ability of our model to predict actions should improve when the already-chosen actions of other agents are provided.

**Encoder** Our model takes as input two modalities that encode the initial scene. The first is the initial state of the agents in the scene which includes the length, width, initial position, initial heading, and object class. We apply a single-layer MLP to encode these values per-agent to an embedding of size $C$. We then add a positional embedding that encodes the agent's order as well as agent identity across the action sequence. The second modality is the map. We use the WOMD representation of a

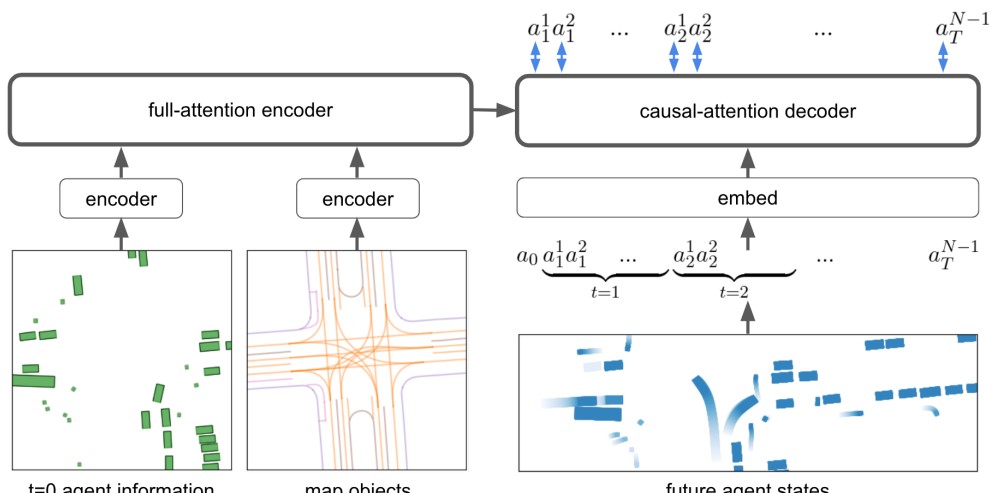

Figure 7: **Trajeglish modeling** We train an encoder-decoder transformer that predicts the action token of an agent conditional on context such as previously selected action tokens, map information, and initial agent states. The diagram visualizes the forward pass of the network during training in which initial agent states and map objects are passed into the network, and the model is trained with standard LLM-like next-token prediction on the sequence of multi-agent action tokens, shown in the top right. The bolded components are transformers.

map as a collection of "map objects", where a map object is a variable-length polyline representing a lane, a sidewalk, or a crosswalk, for example. We apply a VectorNet encoder to encode the map to a sequence of embeddings for at most $M$ map objects (Gao et al., 2020). Note that although the model is not permutation equivariant to the agents, it is permutation invariant to the ordering of the map objects. Similar to Wayformer (Nayakanti et al., 2022), we then apply a layer of latent query attention that outputs a final encoding of the scene initialization.

**Decoder** Given the set of multi-agent future trajectories, we tokenize the trajectories and flatten using the same order used to apply positional embeddings to the $t = 0$ agent encoder to get a sequence $a_0^0 a_1^0 ... a_N^T$. We then prepend a start token and pop the last token, and use an embedding table to encode the result. For timesteps for which an agent's state wasn't observed in the data, we set the embedding to zeros. We pass the full sequence through a transformer with causal mask during training. Finally, we use a linear layer to decode a distribution over the $|V|$ template states and train to maximize the probability of the next token with cross-entropy loss. We tie the token embedding matrix to the weight of the final linear layer, which we observed results in small improvements (Press & Wolf, 2017). We leverage flash attention (Dao et al., 2022) which we find greatly speeds up training time, as documented in Sec. A.8.

We highlight that although the model is trained to predict the next token, it is incorrect to say that a given embedding for the motion token of a given agent only receives supervision signal for the task of predicting the next token. Since the embeddings for later tokens attend to the embeddings of earlier tokens, the embedding at a given timestep receives signal for the task of predicting all future tokens across all agents.

## 4 EXPERIMENTS

We use the Waymo Open Motion Dataset (WOMD) to evaluate Trajeglish in full and partial control environments. We report results for rollouts produced by Trajeglish on the official WOMD Sim Agents Benchmark in Sec. 4.1. We then ablate our design choices in simplified full and partial control settings in Sec. 4.2. Finally, we analyze the representations learned by our model and the density estimates it provides in Sec. 4.3. The hyperparameters for each of the models that we train can be found in Sec. A.4.

### 4.1 WOMD SIM AGENTS BENCHMARK

We test the sampling performance of our model using the WOMD Sim Agents Benchmark and report results in Tab. 1. Submissions to this benchmark are required to submit 32 rollouts of length 8

Table 1: WOMD Sim Agents Test

| Method | Realism meta metric ↑ | Kinematic metrics ↑ | Interactive metrics ↑ | Map-based metrics ↑ | minADE (m) ↓ |
|---|---|---|---|---|---|
| Constant Velocity | 0.2380 | 0.0465 | 0.3372 | 0.3680 | 7.924 |
| Wayformer (Identical) | 0.4250 | 0.3120 | 0.4482 | 0.5620 | 2.498 |
| MTR+++ | 0.4697 | 0.3597 | 0.4929 | 0.6028 | 1.682 |
| Wayformer (Diverse) | 0.4720 | 0.3613 | 0.4935 | 0.6077 | 1.694 |
| Joint-Multipath++ | 0.4888 | 0.4073 | 0.4991 | 0.6018 | 2.052 |
| MTR_E* | 0.4911 | **0.4180** | 0.4905 | 0.6073 | **1.656** |
| MVTA | 0.5091 | **0.4175** | 0.5186 | 0.6374 | 1.870 |
| MVTE* | 0.5168 | **0.4202** | 0.5289 | 0.6486 | 1.677 |
| Trajeglish | **0.5339** | 0.4019 | **0.5811** | **0.6667** | 1.872 |

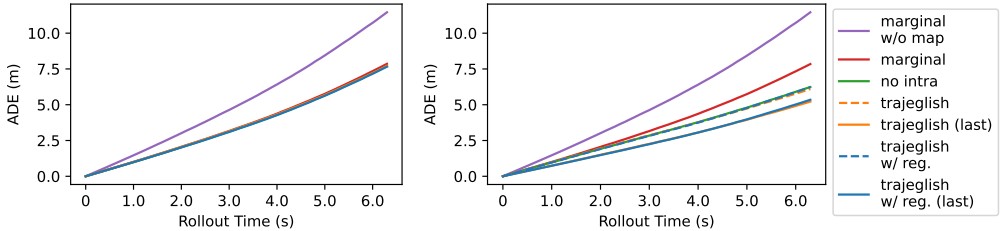

Figure 8: **Partial control ADE** Left shows the ADE for the vehicles selected for evaluation under partial control, but for rollouts where the agents are fully autonomous. Right shows the ADE for the same vehicles but with all other agents on replay. When agents controlled by Trajeglish go first in the permutation order, they behave similarly to the no intra model. When they go last, they utilize the intra-timestep information to produce interaction more similar to recorded logs, achieving a lower ADE.

seconds at 10hz per scenario, each of which contains up to 128 agents. We bold multiple submissions if they are within 1% of each other, as in Montali et al. (2023). Trajeglish is the top submission along the leaderboard meta metric, outperforming several well-established motion prediction models including Wayformer, MultiPath++, and MTR (Shi et al., 2022; 2023), while being the first submission to use discrete sequence modeling. Most of the improvement is due to the fact that Trajeglish models interaction between agents significantly better than prior work, increasing the state-of-the-art along interaction metrics by 9.9%. A full description of how we sample from the model for this benchmark with comparisons on the WOMD validation set is included in Appendix A.5.

## 4.2 ABLATION

To simplify our ablation study, we test models in this section on the scenarios they train on, of at most 24 agents and 6.4 seconds in length. We compare performance across 5 variants of our model. Both "trajeglish" and "trajeglish w/ reg." refer to our model, the latter using the noisy tokenization strategy discussed in Sec. A.2. The "no intra" model is an important baseline designed to mimic the behavior of behavior cloning baselines used in Symphony (Igl et al., 2022) and "Imitation Is Not Enough" (Lu et al., 2023). For this baseline, we keep the same architecture but adjust the masking strategy in the decoder to not attend to actions already chosen for the current timestep. The "marginal" baseline is designed to mimic the behavior of models such as Wayformer (Nayakanti et al., 2022) and MultiPath++ (Varadarajan et al., 2021) that are trained to model the distribution over single-agent trajectories instead of multi-agent scene-consistent trajectories. For this baseline, we keep the same architecture but apply a mask to the decoder that enforces that the model can only attend to previous actions chosen by the current agent. Our final baseline is the same as the marginal baseline but without a map encoder. We use this baseline to understand the extent to which the models rely on the map for traffic modeling.

**Partial control** We report results in Fig. 8 in a partial controllability setting in which a single agent in each scenario is chosen to be controlled by the traffic model and all other agents are set to replay. The single-agent ADE (average distance error) for the controlled-agent is similar in full autonomy rollouts for all models other than the model that does not condition on the map, as expected. How-

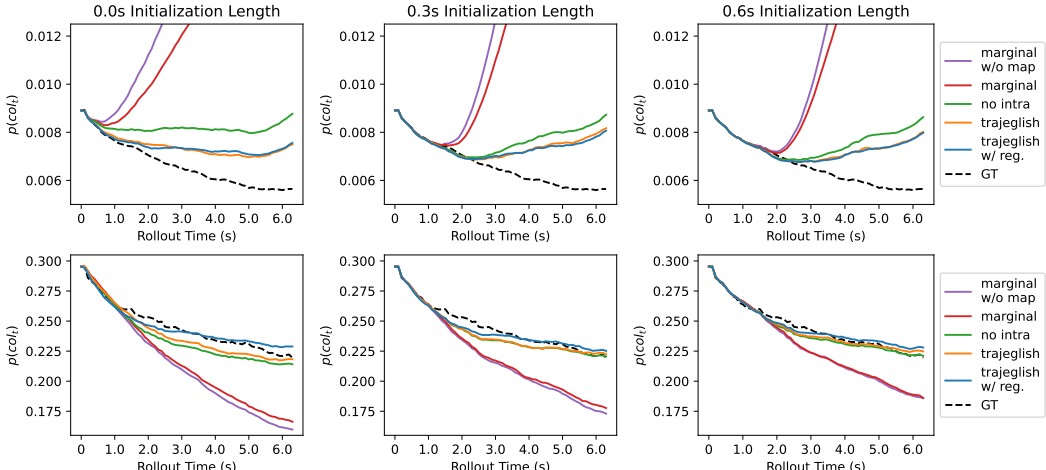

Figure 9: **Full Autonomy Collision Rate** Vehicle collision rate is shown on top and pedestrian collision rate is shown on bottom. From left to right, we seed the scene with an increasing number of initial actions from the recorded data. Trajeglish models the log data statistics significantly better than baselines when seeded with only an initial timestep, as well as with longer initialization.

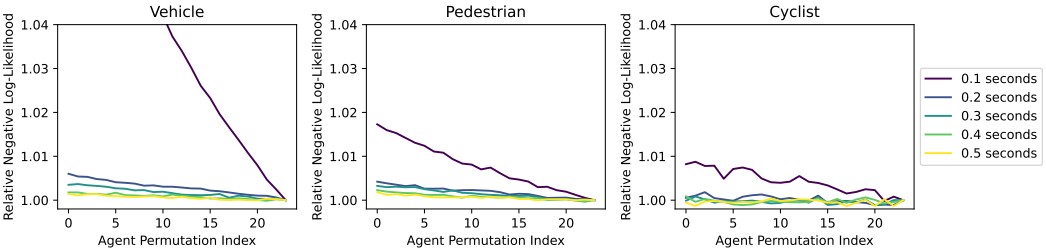

Figure 10: **Intra-Timestep Conditioning** We plot the negative log-likelihood (NLL) when we vary how many agents choose an action before a given agent within a given timestep. As expected, when the context length increases, intra-timestep interaction becomes much less important to take into account.

ever, in rollouts where all other agents are placed on replay, the replay trajectories leak information about the trajectory that the controlled-agent took in the data, and as a result, the no-intra and trajeglish rollouts have a lower ADE. Additionally, the Trajeglish rollouts in which the controlled agent is placed first do not condition on intra-timestep information and therefore behave identically to the no-intra baseline, whereas rollouts where the controlled-agent is placed last in the order provide the model with more information about the replay trajectories and result in a decreased ADE.

**Full control**  We evaluate the collision rate of models under full control in Fig. 9 as a function of initial context, object category, and rollout duration. The value of modeling intra-timestep interaction is most obvious when only a single timestep is used to seed generation, although intra-timestep modeling significantly improves the collision rate in all cases for vehicles. For interaction between pedestrians, Trajeglish is able to capture the grouping behavior effectively. We observe that noising the tokens during training improves rollout performance slightly in the full control setting. We expect these rates to improve quickly given more training data, as suggested by Fig. 11.

## 4.3 ANALYSIS

**Intra-Timestep Dependence**  To understand the extent to which our model leverages intra-timestep dependence, in Fig. 10, we evaluate the negative log likelihood under our model of predicting an agent's next action depending on the agent's order in the selected permutation, as a function of the amount of historical context the model is provided. In all cases, the agent gains predictive power from conditioning on the actions selected by other agents within the same timestep, but the log likelihood levels out as more historical context is provided. Intra-timestep dependence is significantly less important when provided over 4 timesteps of history, as is the setting used for most motion prediction benchmarks.

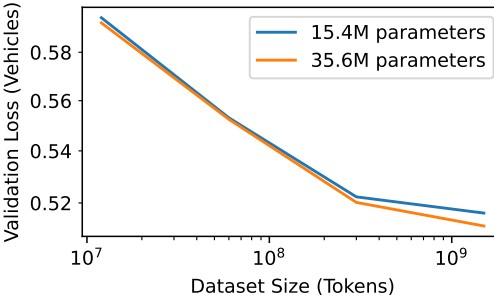 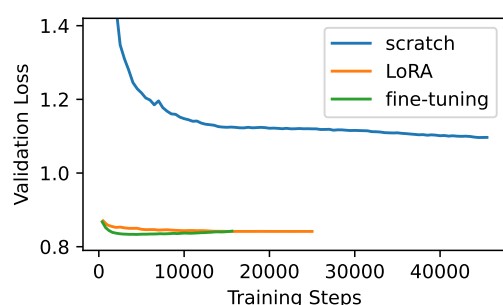

Figure 11: **Scaling Behavior** Our preliminary study on parameter and dataset scaling suggests that, compared to LLMs (Kaplan et al., 2020), Trajeglish is severely data-constrained on WOMD; models with 35M parameters just start to be significantly better than models with 15M parameters for datasets the size of WOMD. A more rigorous study of how all hyperparameters of the training strategy affect sampling performance is reserved for future work.

Figure 12: **nuScenes transfer** We test the ability of our model to transfer to the maps and scenario initializations in the nuScenes dataset. The difference between maps and behaviors found in the nuScenes dataset are such that LoRA does not provide enough expressiveness to fine-tune the model to peak performance. The fine-tuned models both outperform and train faster than the model that is trained exclusively on nuScenes.

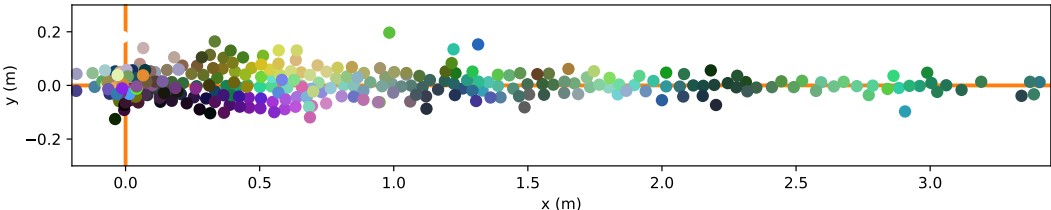

Figure 13: **Token Embedding Visualization** We run PCA on the model embeddings at initialization and at convergence, and plot the $(x, y)$ location of each of the token templates using the top 3 principal component values to determine the hue, lightness, and saturation of the point. The model learns that tokens that correspond to actions close together in euclidean space represent semantically similar actions. Note that the heading of each action is not visualized, which also affects action similarity, especially at low speeds. Additionally, the top 3 principal components include only 35% of the variance, explaining why some colors repeat.

**Representation Transferability**   We measure the generalization of our model to the nuScenes dataset (Caesar et al., 2019). As recorded in Sec. A.8, nuScenes is 3 orders of magnitude smaller than WOMD. Additionally, nuScenes includes scenes from Singapore where the lane convention is opposite that of North America where WOMD is collected. Nevertheless, we show in Fig. 12 that our model can be fine-tuned to a validation NLL far lower than a model trained from scratch on only the nuScenes dataset. At the same time, we find that LoRA (Hu et al., 2021) does not provide enough expressiveness to achieve the same NLL as fine tuning the full model. While bounding boxes have a fairly canonical definition, we note that there are multiple arbitrary choices in the definition of map objects that may inhibit transfer of traffic models to different datasets.

**Token Embeddings**   We visualize the embeddings that the model learns in Fig. 13. Through the task of predicting the next token, the model learns a similarity matrix across tokens that reflects the Euclidean distance between the raw actions that the tokens represent.

**Preliminary Scaling Law**   We perform a preliminary study of how our model scales with increased parameter count and dataset size in Fig. 11. We find that performance between a model of 15.4M parameters and 35.6 parameters is equivalent up to 0.5B tokens, suggesting that a huge amount of performance gain is expected if the dataset size can be expanded beyond the 1B tokens in WOMD. We reserve more extensive studies of model scaling for future work.

## 5   CONCLUSION

In this work, we introduce a discrete autoregressive model of the interaction between road users. By improving the realism of self-driving simulators, we hope to enhance the safety of self-driving systems as they are increasingly deployed into the real world.

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

# A APPENDIX

## A.1 INTRA-TIMESTEP INTERACTION

There are a variety of reasons that intra-timestep dependence may exist in driving log data. To list a few, driving logs are recorded at discrete timesteps and any interaction in the real world between timesteps gives the appearance of coordinated behavior in log data. Additionally, information that is not generally recorded in log data, such as eye contact or turn signals, may lead to intra-timestep dependence. Finally, the fact that log data exists in 10-20 second chunks can result in intra-timestep dependence if there were events before the start of the log data that result in coordination during the recorded scenario. These factors are in general weak, but may give rise to behavior in rare cases that is not possible to model without taking into account coordinatation across agents within a single timestep.

## A.2 REGULARIZATION

Trajeglish is trained with teacher forcing, meaning that it is trained on the tokenized representation of ground-truth trajectories. However, at test time, the model ingests its own actions. Given that the model does not model the ground-truth distribution perfectly, there is an inevitable mismatch between the training and test distributions that can lead to compounding errors (Ross & Bagnell, 2010; Ranzato et al., 2016; Philion, 2019). We combat this effect by noising the input tokens fed as input to the model. More concretely, when tokenizing the input trajectories, instead of choosing the token with minimum corner distance to the ground-truth state as stated in Eq. 3, we sample the token from the distribution

$$a_i \sim \text{softmax}_i(\text{nucleus}(d(\boldsymbol{s}_i, \boldsymbol{s})/\sigma, p_{\text{top}})) \qquad (5)$$

meaning we treat the the distance between the ground-truth raw state and the templates as logits of a categorical distribution with temperature $\sigma$ and apply nucleus sampling (Holtzman et al., 2020) to generate sequences of motion tokens. When $\sigma = 0$ and $p_{\text{top}} = 1$, the approach recovers the tokenization strategy defined in Eq. 3. Intuitively, if two tokens are equidistant from the ground-truth under the average corner distance metric, this approach will sample one of the two tokens with equal probability during training. Note that we retain the minimum-distance template index as the ground-truth target even when noising the input sequence.

While this method of regularization does make the model more robust to errors in its samples at test time, it also adds noise to the observation of the states of other agents which can make the model less responsive to the motion of other agents at test time. As a result, we find that this approach primarily improves performance for the setting where all agents are controlled by the traffic model.

## A.3 TOKENIZATION ANALYSIS

We compare our approach for tokenization against two grid-based tokenizers (van den Oord et al., 2016; Seff et al., 2023; Janner et al., 2021), and one sampling-based tokenizer. The details of these methods are below.

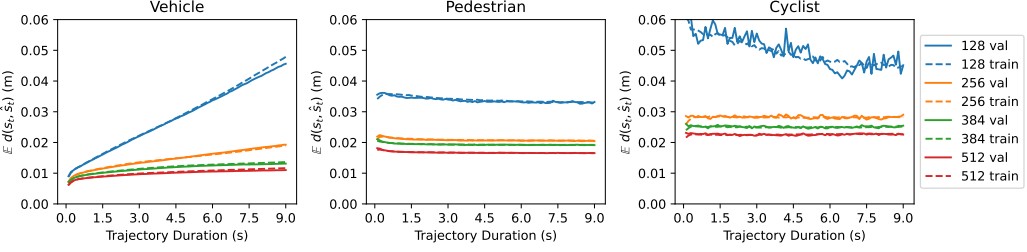

Figure 14: **K-disk expected discretization error** Average corner distance for each of the k-disk vocabularies of sizes 128, 256, 384, and 512.

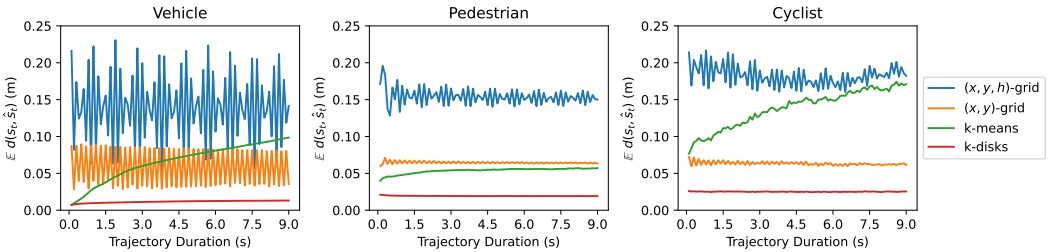

Figure 15: **Tokenization method comparison** Average corner distance for trajectories tokenized with a vocabulary of 384 with template sets derived using different methods.

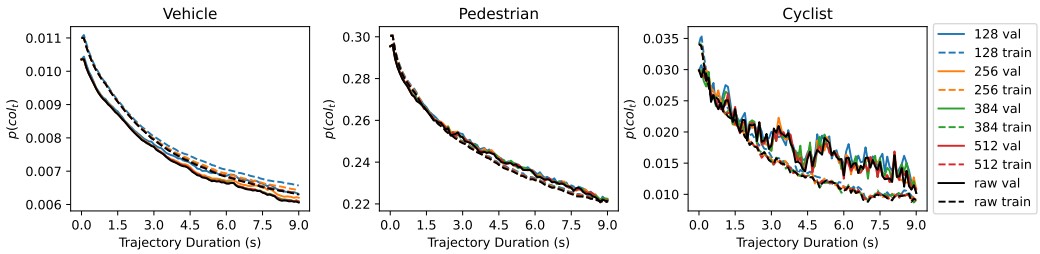

Figure 16: **Semantic Tokenization Performance** We plot the probability that the bounding box of an agent has non-zero overlap with another agent in the scene for each timestep. The collision rate for the raw data is shown in black.

$(x, y, h)$-**grid** - We independently discretize change in longitudinal and lateral position and change in heading, and treat the template set as the product of these three sets. For vocabulary sizes of 128/256/384/512 respectively, we use 6/7/8/9 values for $x$ and $y$, and 4/6/7/8 values for $h$. These values are spaced evenly between (-0.3, 3.5) m for $x$, (-0.2 m, 0.2 m) for $y$, and (-0.1, 0.1) rad for $h$.

$(x, y)$-**grid** - We independently discretize change in only the location. We choose the heading for each template based on the heading of the state-to-state transition found in the data with a change in location closest to the template location. Compared to the $(x, y, h)$-grid baseline, this approach assumes heading is deterministic given location in order to gain resolution in location. We use 12/16/20/23 values for $x$ and $y$ with the same bounds as in the $(x, y, h)$-grid baseline.

**k-means** - We run k-means many times on a dataset of $(x, y, h)$ state-to-state transitions. The distance metric is the distance between the $(x, y)$ locations. We note that the main source of randomness across runs is how k-means is seeded, for which we use k-means++ Arthur & Vassilvitskii (2007). We ultimately select the template set with minimum expected discretization error as measured by the average corner distance.

**k-disks** - As shown in Alg. 1, we sample subsets of a dataset of state-to-state transitions that are at least $\epsilon$ from each other. For vocab sizes of 128/256/384/512, we use $\epsilon$ of 3.5/3.5/3.5/3.0 centimeters.

Intuitively, the issue with both grid-based methods is that they distribute templates evenly instead of focusing them in regions of the support where the most state transitions occur. The main issue with k-means is that the heading is not taken into account when optimizing the cluster centers.

We offer several comparisons between these methods. In Fig. 14, we plot the expected corner distance between trajectories and tokenized trajectories as a function of trajectory length for the template sets found with k-disks. In Fig. 15, we compare the tokenization error as a function of trajectory length and find that grid-based tokenizers create large oscillations. To calibrate to a metric more relevant to the traffic modeling task, we compare the collision rate between raw trajectories as a function of trajectory length for the raw scenarios and the tokenized scenarios using k-disk template sets of size 128, 256, 384, and 512 in Fig. 16. We observe that a vocabulary size of 384 is sufficient to avoid creating extraneous collisions. Finally, Fig. 17 plots the full distribution of discretization

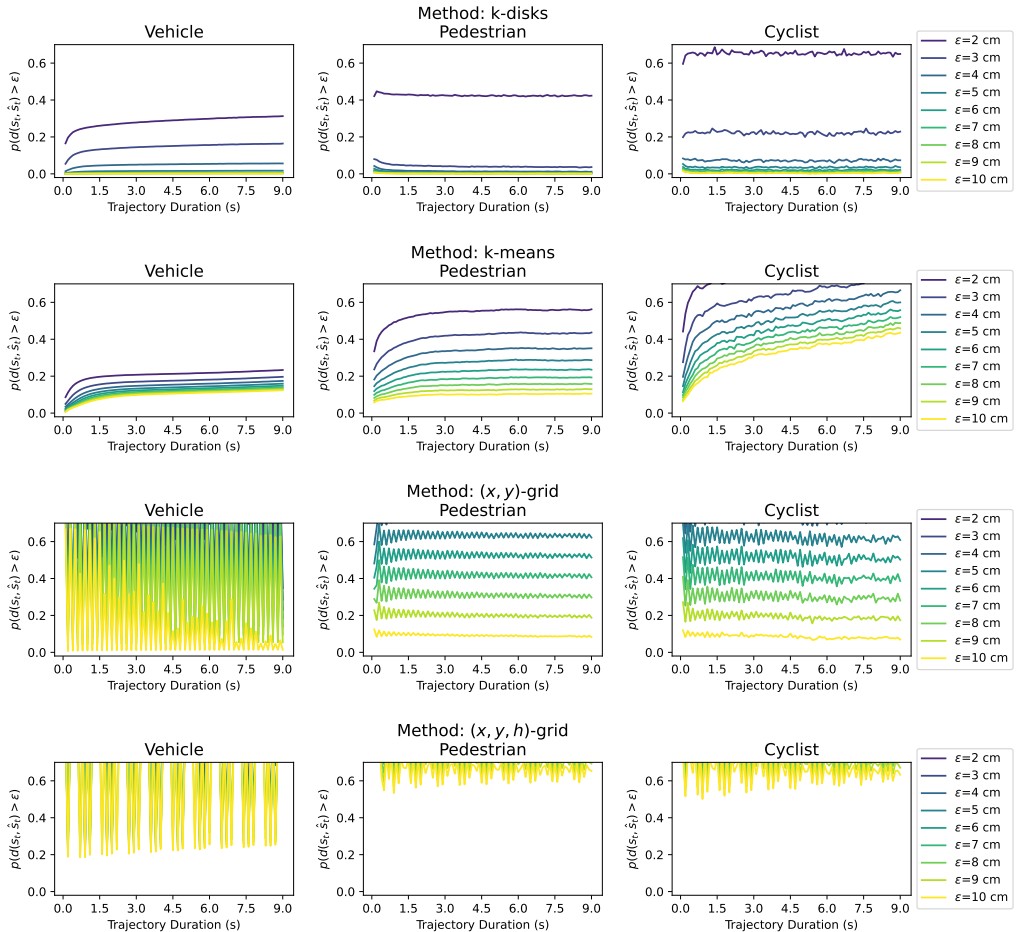

Figure 17: **Discretization error distribution** We plot the probability that the discretized trajectory is greater than 2 cm $\leq \epsilon \leq$ 10 cm away from the true trajectory as a function of trajectory length. Lower is therefore better. Each row visualizes the error distribution for a different method, each with a vocabulary size of 384. We keep the y-axis the same across all plots. We note that k-means discretizes more trajectories to within 2 cm than k-disks, but does not improve past 5 cm, whereas k-disks is able to tokenize nearly all trajectories in WOMD to within 6 centimeters.

errors for each of the baselines and Tab. 2 reports the expected discretization error across vocabulary sizes for each of the methods.

---

**Algorithm 1** Samples a candidate vocabulary of size $N$. The distance $d(x_0, x)$ measures the average corner distance between a box of length 1 meter and width 1 meter with state $x_0$ vs. state $x$.

---

1: **procedure** SAMPLEKDISKS($X$, $N$, $\epsilon$)
2:     $S \leftarrow \{\}$
3:     **while** len($S$) < $N$ **do**
4:         $x_0 \sim X$
5:         $X \leftarrow \{x \in X \mid d(x_0, x) > \epsilon\}$
6:         $S \leftarrow S \cup \{x_0\}$
        **return** $S$

---

Table 2: Tokenization discretization error comparison

| method | $\mathbb{E}[d(s, \hat{s})]$ (cm) | | | |
|---|---|---|---|---|
| | $|V| = 128$ | $|V| = 256$ | $|V| = 384$ | $|V| = 512$ |
| $(x, y, h)$-grid | 20.50 | 16.84 | 14.09 | 12.59 |
| $(x, y)$-grid | 9.35 | 8.71 | 5.93 | 4.74 |
| k-means | 14.49 | 8.17 | 6.13 | 5.65 |
| k-disks | **2.66** | **1.46** | **1.18** | **1.02** |

## A.4 TRAINING HYPERPARAMETERS

We train two variants of our model. The variant we use for the WOMD benchmark is trained on scenarios with up to 24 agents within 60.0 meters of the origin, up to 96 map objects with map points within 100.0 meters of the origin, 2 map encoder layers, 2 transformer encoder layers, 6 transformer decoder layers, a hidden dimension of 512, trained to predict 32 future timesteps for all agents. We train with a batch size of 96, with a tokenization temperature of 0.008, a tokenization nucleus of 0.95, a top learning rate of 5e-4 with 500 step warmup and linear decay over 800k optimization steps with AdamW optimizer (Loshchilov & Hutter, 2017). We use the k-disks tokenizer with vocabulary size 384. During training, we choose a random 4-second subsequence of a WOMD scenario, a random agent state to define the coordinate frame, and a random order in which the agents are fed to the model.

For the models we analyze in all other sections, we use the same setting from above, but train to predict 64 timesteps, using only 700k optimization steps. Training on these longer scenarios enables us to evaluate longer rollouts without the complexity of extending rollouts in a fair way across models, which we do only for the WOMD Sim Agents Benchmark and document in Sec. A.5.

## A.5 EXTENDED ROLLOUTS FOR WOMD SIM AGENTS BENCHMARK

In order to sample scenarios for evaluation on the WOMD sim agents benchmark, we require the ability to sample scenarios with an arbitrary number of agents arbitrarily far from each other for an arbitrary number of future timesteps. While it may become possible to train a high-performing model on 128-agent scenarios on larger datasets, we found that training our model on 24-agent scenarios and then sampling from the model using a "sliding window" (Hu et al., 2023) both spatially and temporally achieved top performance.

In detail, at a given timestep during sampling, we determine the 24-agent subsets with the following approach. First, we compute the 24-agent subset associated with picking each of the agents in the scene to be the center agent. We choose the subset associated with the agent labeled as the self-driving car to be the first chosen subset. Among the agents not included in a subset yet, we find which agent has a 24-agent subset associated to it with the maximum number of agents already included in a chosen subset, and select that agent's subset next. We continue until all agents are included in at least one of the subsets.

Importantly, to define the order for agents within the subset, we place any padding at the front, followed by all agents that will have already selected an action at the current timestep, followed by the remaining agents sorted by distance to the center agent. In keeping this order, we enable the agents to condition on the maximum amount of pre-generated information possible. Additionally, this ordering guarantees that the self-driving car is always the first to select an action at each timestep, in accordance with the guidelines for the WOMD sim agents challenge (Montali et al., 2023).

To sample an arbitrarily long scenario, we have the option to sample up to $t < T = 32$ steps before computing new 24-agent subsets. Computing new subsets every timestep ensures that the agents within a subset are always close to each other, but has the computational downside of requiring the transformer decoder key-value cache to be flushed at each timestep. For our submission, we compute the subsets at timesteps $t \in \{10, 34, 58\}$.

Table 3: 2023 WOMD sim agents validation

| Method | Realism Meta metric ↑ | Kinematic metrics ↑ | Interactive metrics ↑ | Map-based metrics ↑ |
|---|---|---|---|---|
| $\tau = 1.25$, $p_{\text{top}} = 0.995$ | 0.5176 | 0.3960 | 0.5520 | 0.6532 |
| $\tau = 1.5$, $p_{\text{top}} = 1.0$ | 0.5312 | 0.3963 | 0.5838 | 0.6607 |
| $\tau = 1.5$, $p_{\text{top}} = 1.0$, w/ $h$-smooth | 0.5352 | 0.4065 | 0.5841 | 0.6612 |

Table 4: WOMD sim agents validation - updated metrics

| Method | Realism Meta metric ↑ | Kinematic metrics ↑ | Interactive metrics ↑ | Map-based metrics ↑ |
|---|---|---|---|---|
| Trajeglish ($\tau = 1.5$) | 0.6078 | 0.4019 | 0.7274 | 0.7682 |
| MTR_E | 0.6348 | 0.4180 | 0.7416 | **0.8400** |
| MVTA | 0.6361 | 0.4175 | 0.7543 | 0.8253 |
| Trajeglish ($\tau = 1.0$) | 0.6437 | 0.4157 | 0.7816 | 0.8213 |
| MVTE | 0.6448 | **0.4202** | 0.7666 | 0.8387 |
| Trajeglish ($\tau = 1.0$, AA=32) | **0.6451** | 0.4166 | **0.7845** | 0.8216 |

While the performance of our model under the WOMD sim agents metrics was largely unaffected by the choice of the hyperparameters above, we found that the metrics were sensitive to the temperature and nucleus that we use when sampling from the model. We use a temperature of 1.5 and a nucleus of 1.0 to achieve the results in Tab. 1. Our intuition for why larger temperatures resulted in larger values for the sim agents metric is that the log likelihood penalizes lack of coverage much more strongly than lack of calibration, and higher temperature greatly improves the coverage.

Finally, we observed that, although the performance of our model sampling with temperature 1.5 was better than all prior work for interaction and map-based metrics as reported in Tab. 3, the performance was worse than prior work along kinematics metrics. To test if this discrepancy was a byproduct of discretization, we trained a "heading smoother" by tokenizing trajectories, then training a small autoregressive transformer to predict back the original heading given the tokenized trajectory. On tokenized ground-truth trajectories, the heading smoother improves heading error from 0.58 degrees to 0.33 degrees. Note that the autoregressive design of the smoother ensures that it does not violate the closed-loop requirement for the Sim Agents Benchmark. The addition of this smoother did improve along kinematics metrics slightly, as reported in Tab. 3. We reserve a more rigorous study of how to best improve the kinematic realism of trajectories sampled from discrete sequence models for future work.

A.6 DECEMBER 28, 2023 - UPDATED SIM AGENTS METRICS

On December 28, 2023, Waymo announced an adjustment to the metrics for the Sim Agents benchmark to improve accuracy of vehicle and off-road collision checking (more details about this adjustment can be found here). Upon re-optimizing hyperparameters of Trajeglish for the new metrics, we found that the optimal sampling hyperparameters were $\tau = 1.0$ and $p_{\text{top}} = 1.0$, which is more intuitive than our previously chosen hyperparameter of $\tau = 1.5$ given that the metrics are intended to measure the extent to which the distribution of sampled scenarios and recorded scenarios match. We then re-trained our model to condition on 32 agents at a time instead of 24 which also improved results slightly. For the final leaderboard results before the announcement of the 2024 Sim Agents Challenge, Trajeglish did end up ahead of all models it had beaten under the previous metrics, although by much slimmer margins, shown in Tab. 4.

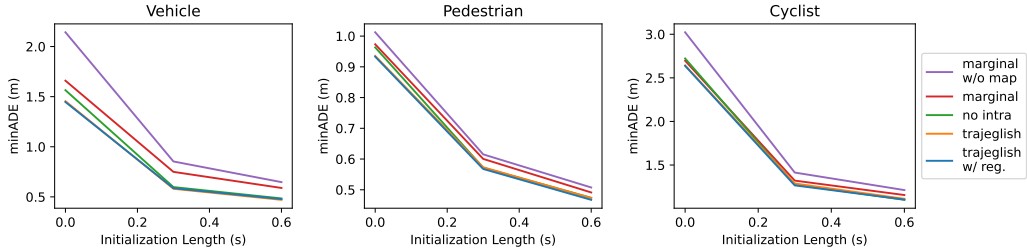

Figure 18: **Full Autonomy minADE** As we seed the scene with a longer initialization, the no-intra model and our model converge to similar values, and all models improve. When initialized with only a single timestep, the performance gap between models that take into account intra-timestep interaction and models that do not is significant.

## A.7 ADDITIONAL ABLATION RESULTS

**Full Control** In Fig. 18, we find the sampled scenario with minimum corner distance to the ground-truth scenario and plot that distance as a function of the number of timesteps that are provided at initialization. When the initialization is a single timestep, the minADE of both models that take into account intra-timestep dependence improves. As more timesteps are provided, the effect diminishes, as expected. We visualize a small number of rollouts in the full autonomy setting in Fig. 21, and videos of other rollouts can be found on our project page.

**Partial Control** To quantitatively evaluate these rollouts, we measure the collision rate and visualize the results in Fig. 19. Of course, we expect the collision rate to be high in these scenarios since most of the agents in the scene are on replay. For Trajeglish models, we find that when the autonomous agent is the first in the permutation to choose an action, they reproduce the performance of the model with no intra-timestep dependence. When the agent goes last however, the collision rate drops significantly. Modeling intra-timestep interaction is a promising way to enable more realistic simulation with some agents on replay, which may have practical benefits given that the computational burden of simulating agents with replay is minimal. In Fig. 22, we visualize how the trajectory for agents controlled by Trajeglish shifts between the full autonomy setting and the partial autonomy setting. The agent follows traffic flow and cedes the right of way when replay agents ignore the actions of the agent controlled by the traffic model.

## A.8 ADDITIONAL ANALYSIS

**Data and Training Statistics** We report a comparison between the number of tokens in WOMD and the number of tokens in datasets used to train GPT-1 and GPT-2 in Tab. 5. Of course, a text token and a motion token do not have exactly the same information content, but we still think the comparison is worth making as it suggests that WOMD represents a dataset size similar to BookCorpus Zhu et al. (2015) which was used to train GPT-1 and the scaling curves we compute for our model shown in Fig. 11 support this comparison. We also report the number of tokens collected per hour of driving to estimate how many hours of driving would be necessary to reach a given token count. In Tab. 6, we document the extent to which using mixed precision and flash attention improves memory use and speed. Using these tools, our model takes 2 days to train on 4 A100s.

**Context Length** Context length refers to the number of tokens that the model has to condition on when predicting the distribution over the next token. Intuitively, as the model is given more context, the model should get strictly better at predicting the next token. We quantify this effect in Fig. 20. We find that the relative decrease in cross entropy from increasing the context length drops off steeply for our model for pedestrians and cyclists, which aligns with the standard intuition that these kinds of agents are more Markovian. In comparison, we find a significant decrease in cross entropy with up to 2 seconds of context for vehicles, which is double the standard context length used for vehicles on motion prediction benchmarks (Ettinger et al., 2021; Caesar et al., 2019).

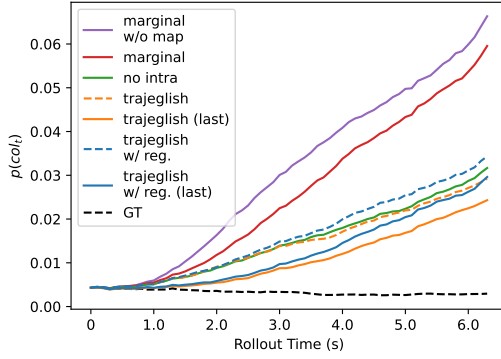

Figure 19: **Partial control collision rate** We plot the collision rate as a function of rollout time when the traffic model controls only one agent while the rest are on replay. We expect this collision rate to be higher than the log collision rate since the replay agents do not react to the dynamic agents. We note that the collision rate decreases significantly just by placing the agent last in the order, showing that the model learns to condition on the actions of other agents within a single timestep effectively.

Figure 20: **Context Length** We plot the negative log-likelihood (NLL) when we vary the context length at test-time relative to the NLL at full context. Matching with intuition, while pedestrians and cyclists are more Markovian on a short horizon, interaction occurs on a longer timescale for vehicles.

Table 5: Dataset comparison by tokens

|  | tokens | rate (tok/hour) |
| --- | --- | --- |
| nuScenes | 3M | 0.85M |
| WOMD | 1.5B | 1.2M |
| WOMD (moving) | 1.1B | 0.88M |
| BookCorpus (GPT-1) | 1B | - |
| OpenWebText (GPT-2) | 9B | - |

Table 6: Training efficiency

|  | memory | speed (steps/hour) |
| --- | --- | --- |
| no intra | 14.7 MiB | 8.9k |
| Trajeglish (mem-efficient) | 7.2 MiB | 11.1k |
| Trajeglish (bfloat16+flash) | 5.6 MiB | 23.0k |

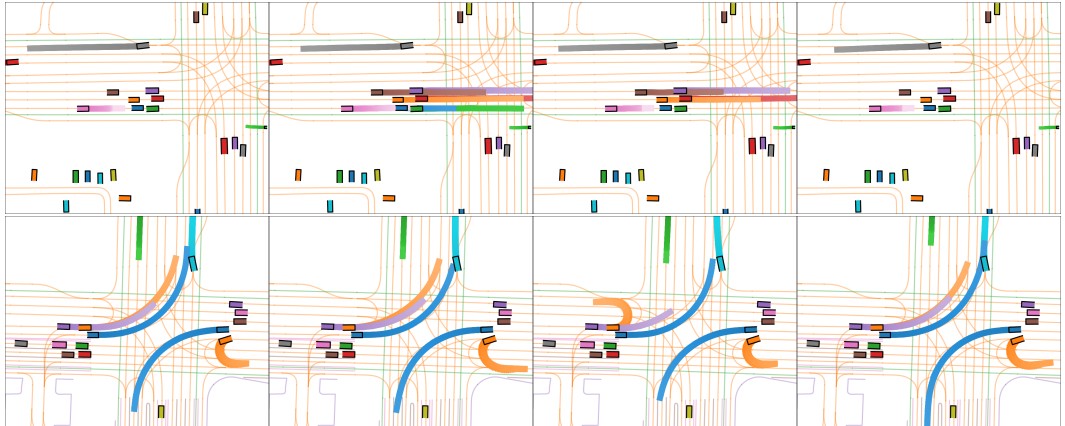

Figure 21: **Full control rollouts** Additional visualizations of full control samples from our model. The model captures the collective behavior of agents at an intersection and individual maneuvers such as U-turns.

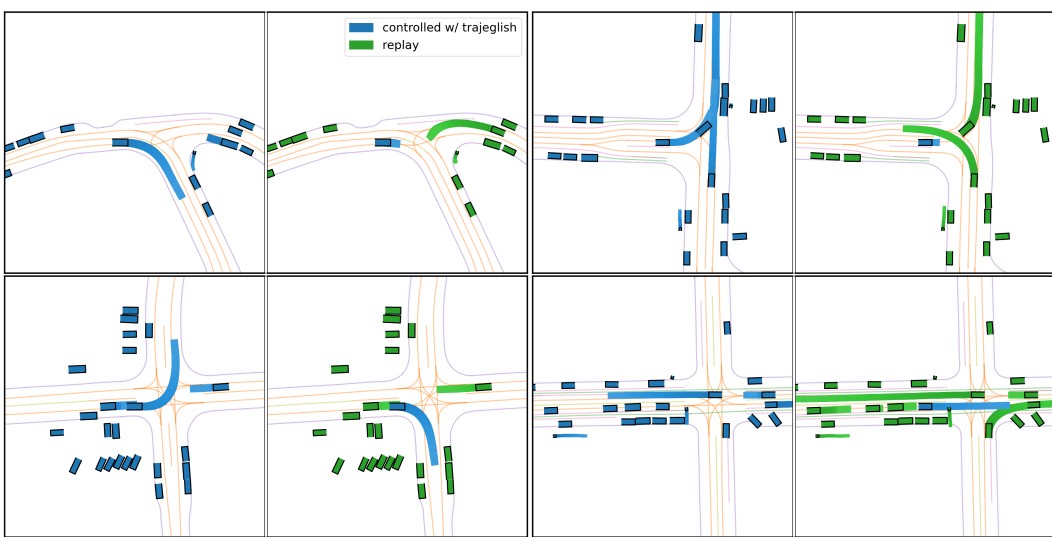

Figure 22: **Partial control comparison** We visualize the effect of controlling only one agent with Trajeglish and controlling the others with replay. The left scene in each pair is a full control sample from Trajeglish. The right scene is generated by placing all green cars on fixed replay tracks and controlling the single blue car with Trajeglish. Our model reacts dynamically to other agents in the scene at each timestep.

