# OpenReview forum: "Trajeglish: Traffic Modeling as Next-Token Prediction"
_ICLR.cc/2024/Conference — ICLR 2024 poster_

### Official Review · Reviewer_mABk · 2023-10-17

**Soundness:** 3 good
**Presentation:** 3 good
**Contribution:** 2 fair
**Rating:** 6
**Confidence:** 3

**Summary:**

This paper proposes a method called Trajeglish to generate future trajectories for traffic participants in a scenario. In particular, they propose a method to tokenize trajectory data using a small vocabulary. Besides, they propose a transformer-based architecture for modeling the action tokens on map information as well as initial states of traffic participants. To evaluate Trajeglish, the authors compare it with a behavior cloning method and a baseline that only models single agent trajectories. The result shows that their method achieves superior performance.

**Strengths:**

### 1.The idea of tokenization using a small vocabulary is moderately novel.

### 2.The visualization and illustration are well made and help the readers to understand the paper.

**Weaknesses:**

## Major:

### 1.motivation of using tokenization (compared with using the actual values as in most of existing work in Appendix B) is not very clear.

### 2.the experimental results are not very impressive

(1) Improvements in Table 1 seem quite small. Can you show standard deviations for the results?

(2) only evaluate on open-loop simulation but not on close-loop simulation

(3) the baseline details are not given (e.g., “The “marginal” baseline is an equally important baseline designed to mimic the behavior of models such as Wayformer (Nayakanti et al., 2022) and MultiPath++ (Varadarajan et al., 2021) that are trained to model the distribution over single agent trajectories instead of multi-agent scene-consistent trajectories.” However, it is unclear if this baseline really can achieve similar performance as Wayformer / MultiPath++ as the authors did not give further details) and it is hard for one to assess if they are really strong baselines.

### 3.motivation of having a model that take order into account is not very convincing

In particular, the idea of having this order seems very unnatural. For example, in the real world the likelihood of equally capable drivers (whether human or AI) to have collisions should be equal?

## Minor:

### 1.missing some relevant work on multi-agent trajectory prediction:

Hivt: Hierarchical vector transformer for multiagent motion prediction, Z. Zhou, L. Ye, J. Wang, K. Wu, and K. Lu.

Language-Guided Traffic Simulation via Scene-Level Diffusion, Z. Zhong, D. Rempe, Y. Chen, B. Ivanovic, Y. Cao, D. Xu, M. Pavone, B. Ray

### 2.did not discuss the limitations of the current work

**Questions:**

-What’s the motivation of using a small vocabulary compared with using the actual values as in most of existing work (as in Appendix B)?

-The provided video is a bit confusing. How do you control other vehicles that are neither replay nor trajeglish? In some videos the legends only show these two types but there are vehicles of other colors.

-Can you also show the variance for Figure 8?

-Figure 9 why the collision rate decreases when the rollout becomes longer?

-Trajgelish behavior under longer horizon rollout (e.g., 200 timesteps)?

-For the experiments, 16 scenarios are sampled for every clip in WOMD? Or only 16 clips in total?

-Can you also provide some qualitative visualization for nuscenes in Appendix?

---

> ### Author Response · Authors · 2023-11-18
> **Thank you for the review (Part 1)**
>
> We thank the reviewer for their questions and feedback. We are glad the reviewer appreciates the novelty of our method for tokenizing trajectories as well as the quality of the visualizations we’ve created to communicate how our method works. We hope to answer the reviewer’s questions and clear up the reviewer’s misunderstandings about our method with our response below:
>
> 2 misunderstandings to clarify:
> - “only evaluate on open-loop simulation but not on close-loop simulation” - we want to emphasize that we exclusively evaluate in closed-loop simulation in this paper. We consider a key component of our contribution to be that we identify formal requirements for a tokenization strategy to enable closed-loop policies (Section 2) and then introduce a tokenization method that satisfies these requirements (Section 3). For the full autonomy evaluation in Figure 9 and Table 1, the partial autonomy evaluation in Figure 10, and our results on the WOMD sim agents benchmark [(see our note to all reviewers)](https://openreview.net/forum?id=Z59Rb5bPPP&noteId=VvSB6yXfHY), the rollouts are fully closed-loop; the encoder-decoder transformer functions as a policy that outputs a single action representing the change in state over the next 0.1 seconds for each agent at each step, upon observing all previously selected actions for all agents.
> - “The provided video is a bit confusing. How do you control other vehicles that are neither replay nor trajeglish?” - All agents in our visualization and evaluation are either controlled by replay or trajeglish. For the left 3 panes of the video, we control all vehicles in the scene with trajeglish and title these panes “Full Autonomy '' to indicate the setting. These are the rollouts we use to evaluate our method in Table 1, Figure 9, and the WOMD sim agents benchmark (see our note to all reviewers). For the rightmost pane, only the blue agent is controlled by trajeglish and the rest are on replay, as we denote in the legend. These are the rollouts we use to evaluate our method in Figure 10.
>
>     To be clear, our ultimate goal is to control agents in simulation such that they interact in realistic ways to any number of agents controlled by a black-box AV system. In the absence of a black-box AV system at hand to test, we use replay as a surrogate black-box controller. The purpose of our experiments in the "partial control setting" is to demonstrate the property of trajeglish agents that they interact with other agents in a closed-loop setting independent of the underlying method that controls the other agents.

---

> ### Author Response · Authors · 2023-11-18
> **Thank you for the review (Part 2)**
>
> Responses to the reviewer’s other questions and requests:
> - “motivation of using tokenization (compared with using the actual values as in most of existing work in Appendix B) is not very clear” + “What’s the motivation of using a small vocabulary compared with using the actual values as in most of existing work (as in Appendix B)?” - Our motivation stems from the impressive results we’ve seen from applying discrete sequence models in other continuous domains, for instance Paella [1], Stable Diffusion [2] Parti [3], and GAIA-1 [4]. We share the intuition behind these works that sacrificing a small amount of resolution (e.g. a few centimeters in our case as reported in Figure 12 and Figure 13) can be well worth the expressiveness and stability of the categorical distribution compared to a mixture-of-x parameterization, for instance. Empirically, this tradeoff appears to be well worth exploring, as our method outperforms all prior work based on diffusion, VAEs, and mixture-of-x modeling on the WOMD sim agents benchmark [(see our note above)](https://openreview.net/forum?id=Z59Rb5bPPP&noteId=VvSB6yXfHY).
> - “Improvements in Table 1 seem quite small. Can you show standard deviations for the results?” We would like to highlight that for the mADE and ADE reported in Table 1, the increase in performance due to intra-timestep conditioning and tokenization regularization is quite significant, improving by 6% and 2.3% respectively. We additionally quantify the performance gap due to these modeling choices in Figure 9 and Figure 10 which show significant improvements when intra-timestep coupling is taken into account. Finally, trajeglish improves on interaction metrics by 4.4% over all prior work on the official WOMD sim agents benchmark, a large jump in performance over all prior work [(see our note above)](https://openreview.net/forum?id=Z59Rb5bPPP&noteId=VvSB6yXfHY).
> - “it is unclear if this baseline really can achieve similar performance as Wayformer / MultiPath++ as the authors did not give further details) and it is hard for one to assess if they are really strong baselines” - In [our note above](https://openreview.net/forum?id=Z59Rb5bPPP&noteId=VvSB6yXfHY), we report results on the WOMD sim agents benchmark which features an official submission for Wayformer as well as a submission based on MultiPath++. Trajeglish improves over Wayformer by 9.2%, and MultiPath++ by 5.4%. We hope these direct comparisons are convincing for the reviewer. We are also updating our draft to include more details about our implementation of the “no-intra” and “marginal” baselines referenced in Table 1.
> - “motivation of having a model that take order into account is not very convincing” - We note that the focus of our work is on how we apply discrete sequence modeling to multi-agent trajectories, and less about the design of our encoder for scene context. We are aware of prior work that studies how to adjust the attention mechanism to maintain permutation equivariance while still leveraging tracking information, such as AgentFormer [5]. However, we find we are able to achieve state of the art performance for traffic modeling by simply randomizing the order during training, so we reserve the improvement of our architecture for encoding scene context for later work.
>
> [1] “A Novel Sampling Scheme for Text- and Image-Conditional Image Synthesis in Quantized Latent Spaces”, Rampas et al, 2023.\
> [2] “High-Resolution Image Synthesis with Latent Diffusion Models”, Rombach et al, 2021.\
> [3] “Scaling Autoregressive Models for Content-Rich Text-to-Image Generation”, Yu et al, 2022.\
> [4] “GAIA-1: A Generative World Model for Autonomous Driving”, Hu et al, 2023.\
> [5] “AgentFormer: Agent-Aware Transformers for Socio-Temporal Multi-Agent Forecasting”, Yuan et al, 2021.

---

> ### Author Response · Authors · 2023-11-18
> **Thank you for the review (Part 3)**
>
> - Additional papers to cite - We note that HiVT studies how mixture-of-x distributions can be used to model single-agent trajectories and “Language-Guided Traffic Simulation via Scene-Level Diffusion” focuses on how text can be used to control generation of a static traffic scenario, so we don’t consider them directly relevant to our work which studies how discrete sequence models can be applied to closed-loop traffic simulation. However, if the reviewer feels strongly that these are relevant, we can add these citations.
> - “did not discuss the limitations of the current work” - we discuss the fact that our encoder-decoder transformer is not affine equivariant and not permutation equivariant in section 3.2 and the fact that map representation may be a limiting factor for successful transfer to other datasets in section 4.3. We will elaborate on these limitations and update our paper.
> - “Can you also show the variance for Figure 8?” - Yes, we will update our paper before the end of the discussion period.
> - “Figure 9 why the collision rate decreases when the rollout becomes longer?” - the reason is that for the rollouts evaluated in Figure 9, we adjust the padding of the WOMD scenarios to prevent agents from reappearing if they ever disappear. As a result, the number of agents in the scene decreases over time, so the number of collisions also decreases over time. We use this setting for evaluation because our tokenizer relies on the existence of the previous state in order to tokenize, so we are unable to tokenize states that do not have a recorded state in the timestep before. On the other hand, for the WOMD sim agents benchmark [(see our note above)](https://openreview.net/forum?id=Z59Rb5bPPP&noteId=VvSB6yXfHY), we evaluate in the setting where all agents visible at the tenth timestep need to be simulated for all 80 timesteps into the future, as required by the benchmark. We will add these details on how padding is taken into account in each of our evaluation settings to our paper.
> - “Trajgelish behavior under longer horizon rollout (e.g., 200 timesteps)?” We will add visualizations of longer rollouts to our video.
> - “For the experiments, 16 scenarios are sampled for every clip in WOMD? Or only 16 clips in total?” 16 scenarios are sampled per clip for each of the 44k WOMD validation clips.
> - “Can you also provide some qualitative visualization for nuscenes in Appendix?” Yes, we will update our paper before the end of the discussion period.

---

> > ### Comment · Reviewer_mABk · 2023-11-21
> > **Thank you for the response!**
> >
> > Thank you for the response! The responses have addressed my major concerns. I will update my score once the updated version is available.

---

### Official Review · Reviewer_qzEz · 2023-10-27

**Soundness:** 3 good
**Presentation:** 3 good
**Contribution:** 2 fair
**Rating:** 6
**Confidence:** 4

**Summary:**

“Trajeglish: Learning the Language of Driving Scenarios” proposes a model that can create scene-consistent rollouts for a subset of agents in a scene. In particular, the proposal consists of a tokenization algorithm, “k-disks”, for tokenization an agent’s motion, and a transformer-based model architecture that autoregressively and causally rolls out agents’ future trajectories. The authors provide competitive results on WOMD and transfer to nuScenes.

**Strengths:**

* Strong tokenizer k-disks outperforming kMeans baselines with low discretization errors and convincing ablation study
* Autoregressive and casual rollouts
* Experiments demonstrating the benefits of intra-timestep dependence of agents
* Experiments demonstrating the transfer to nuScenes

**Weaknesses:**

* Missing WOMD baseline results from other models
* Similar contributions as the recently published “MotionLM: Multi-Agent Motion Forecasting as Language Modeling” (https://arxiv.org/pdf/2309.16534.pdf)

**Questions:**

* How does your approach compare to the recently published “MotionLM: Multi-Agent Motion Forecasting as Language Modeling” (https://arxiv.org/pdf/2309.16534.pdf)?

---

> ### Author Response · Authors · 2023-11-18
> **Thank you for the review**
>
> We thank the reviewer for their valuable feedback. We are glad that the reviewer appreciates several aspects of our work, including the “strong tokenizer k-disks” performance with “convincing ablation study” of different tokenization approaches as well as experiments investigating “the benefits of [modeling] intra-timestep dependence of agents”, and our “experiments demonstrating the transfer to nuScenes”. We respond to the reviewer’s comments below:
>
> - “Missing WOMD baseline results from other models” - In order to offer direct comparison to baseline results from other models, we have made a submission to the WOMD sim agents benchmark, and documented the results in [our note above to all reviewers](https://openreview.net/forum?id=Z59Rb5bPPP&noteId=VvSB6yXfHY). To summarize, trajeglish outperforms all other single-model submissions, including models based on the Wayformer, Multipath++, and MTR architectures, and is in 2nd place, falling just behind a submission that uses ensembling by only 0.25%. Our method additionally sets a new state of the art for the interaction metrics measured by the benchmark, surpassing all prior work by 4.4%. We hope these results provide the comparison to WOMD baseline results that the reviewer was hoping to see for our method.
> - "Similar contributions as the recently published MotionLM" + “How does your approach compare to MotionLM?” - The main difference between our work and MotionLM is the application they target; MotionLM targets online deployment, and therefore models 1- and 2-agent (x,y) trajectories at 2hz. Since we target offline simulation of agents, we model up to 24-agent (x,y,h) trajectories at 10hz. The differences between the distributions that these methods seek to model give rise to differences in tokenization schemes; MotionLM discretizes the second-order derivatives of motion in a global coordinate frame, whereas we leverage the fact that we decode heading to discretize first-order derivatives of motion in the per-timestep Frenet frame of the agent. As a practical consideration, the difference in target distribution also results in significantly different sequence lengths modeled between the two approaches; sequences modeled by MotionLM are of length 2 agents * 16 timesteps = 32, while sequences modeled by trajeglish are of length 24 agents * 63 timesteps = 1512. Finally, while temporal causality is shown to be a useful inductive bias for MotionLM and the attention mask used by the autoregressive decoder does not take into account intra-timestep conditioning, our goal in Section 2 is to show that temporal causality is actually required for our application of controlling reactive agents in simulation, and we explicitly measure in Figures 9, 10, and 11 to what extent intra-timestep conditioning affects performance, especially when only a single timestep of trajectory context is provided at initialization.
>
>     While we plan to add a citation for MotionLM to our paper, we would also like to make it clear that MotionLM was released on arxiv on September 28, the same day that we submitted our work to ICLR. As a result, we hope that the reviewer recognizes our work and MotionLM as fully concurrent successful applications of discrete sequence modeling to two different variants of motion prediction.

---

> > ### Comment · Reviewer_qzEz · 2023-11-21
> > **Thank you for the response**
> >
> > Thank you for the detailed response.
> >
> > **WOMD results**: Thank you for adding the new results on the WOMD sim agents benchmark. I'm happy to see this comparison to the baselines, which makes your work stronger.
> >
> > **MotionLM**: Thanks a lot for the detailed comparison to MotionLM. It's great that you will add a reference to this publication. I also agree that both publications should be regarded as concurrent for the review.
> >
> > I have no additional concerns and am looking forward to the updated version.

---

### Official Review · Reviewer_KYot · 2023-11-02

**Soundness:** 3 good
**Presentation:** 3 good
**Contribution:** 2 fair
**Rating:** 6
**Confidence:** 4

**Summary:**

This paper presents a language modeling inspired approach to data-driven traffic simulation. The key step involved is tokenizing future driving scenario data into a sequential, language-style format, for which this paper compares several tokenization schemes. Once tokenized, a simple transformer encoder-decoder architecture is proposed to encode the initial scene state and autoregressively decode the tokenized scene future. Training the model follows the standard next token prediction objective as in language modeling, with an optional noise term on the ground truth tokens to deal with distribution shifts caused by teacher forcing. A diverse set of experiments on the Waymo Open Motion Dataset (WOMD) provide several insights on how design choices for this new formulation impact simulation quality.

**Strengths:**

The key strengths of this work lie in its conceptual and architectural simplicity in comparison to existing methods. The idea is well-motivated and the presentation is clear. Besides this, the paper provides a detailed experimental analysis on different aspects of the proposed design space.

**Weaknesses:**

1. The benchmarking in Table 1 follows a much simpler setting with fewer max agents (24 vs. 128) and a shorter time horizon (6 seconds vs. 8 seconds) than prior work on WOMD [1,2,3].
2. As a result of this simpler benchmark and missing comparisons to any prior architecture, this paper does not address the key question of whether the proposed method is competitive to the current state-of-the-art despite its simplicity. At a glance, it seems to be much worse, with a minADE >3m in comparison to the SoTA methods with minADE < 1m on the more challenging standard WOMD setting.
3. The paper is not self-contained, with important details (e.g., related work and several figures referenced during discussions in the main paper) only available in the appendix

[1] https://arxiv.org/abs/2209.13508

[2] https://arxiv.org/abs/2306.17770

[3] https://arxiv.org/abs/2309.16534

**Questions:**

1. Please see “Weaknesses” - these are the key points with the most influence on my rating. If addressed via a fair and direct comparison to existing work, I am inclined to improve my rating.
2. Given the simple and scalable architecture, it would be interesting to analyze the importance of scale (in terms of #parameters in the encoder/decoder) towards the performance of the proposed model.
3. The clarity of Figure/Table captions and their placement within the document could be improved, currently, they are often very far from the text referencing them.
4. How are actors ordered in the decoder? Is this randomized for each scene during both training and inference?

---

> ### Author Response · Authors · 2023-11-18
> **Thank you for the review**
>
> We thank the reviewer for their valuable feedback. We are glad that the reviewer found our method “well-motivated” and our presentation “clear”, with “detailed experimental analysis on different aspects of the proposed design space”. We respond to the reviewer’s comments below:
>
> - “This paper does not address the key question of whether the proposed method is competitive to the current state-of-the-art despite its simplicity” + “If addressed via a fair and direct comparison to existing work, I am inclined to improve my rating.”  - As outlined in our [note to all reviewers](https://openreview.net/forum?id=Z59Rb5bPPP&noteId=VvSB6yXfHY) above, to compare more directly to prior work, we have prepared a submission to the Waymo “sim agents” leaderboard. To summarize the results, trajeglish is the top performing method among single-model submissions, surpassing methods built on top performing architectures for motion prediction such as Wayformer, Multipath++, and MTR. We are adding the details to our paper of how we use our model to rollout for all 128 agents for the full 80 timesteps as required for submissions to the benchmark. We hope these results provide the direct comparison the reviewer is hoping to see for our method.
> - “At a glance, it seems to be much worse, with a minADE >3m in comparison to the SoTA methods with minADE < 1m” - The minADE in Table 1 is calculated by sampling scenarios given only 1 timestep of ground-truth history. We expect the optimal minADE for this setting to be significantly higher than the optimal minADE when 11 timesteps are given as input, as is the case for all 3 WOMD benchmarks related to motion prediction. For an apples-to-apples comparison of minADE to prior work, we report a minADE of 1.68 meters on the WOMD sim agents benchmark (see our [note to all reviewers](https://openreview.net/forum?id=Z59Rb5bPPP&noteId=VvSB6yXfHY)), which is competitive with that of other methods, although we did not optimize hyperparameters for this particular metric since it is not included in the benchmark’s ranking metric. In the paper, we will be more precise about the evaluation setting we use for Table 1.
> - “The paper is not self-contained, with important details (e.g., related work and several figures referenced during discussions in the main paper) only available in the appendix” + “The clarity of Figure/Table captions and their placement within the document could be improved” - We appreciate this feedback. We are reorganizing the paper to make sure the related work and main figures are included in the body of the paper. We will also edit the captions for clarity, and post an updated version before the end of the discussion period.
> - “it would be interesting to analyze the importance of scale (in terms of #parameters in the encoder/decoder) towards the performance of the proposed model” - we certainly agree with the sentiment of this proposal. The fact that discrete sequence modeling has proven to be so reliably scalable in other domains is one of our main motivations for studying how these models can be applied to traffic modeling. We will add a limited study of scaling laws to the paper before the end of the discussion period.
> - “How are actors ordered in the decoder? Is this randomized for each scene during both training and inference?” - During training, we randomize the order. As a result, we are free to choose any order at inference. We use this capability to evaluate how predictive ability changes as we adjust the agent's position in the ordering, as shown in Figure 11. Across all other experiments, we keep the order stable by sorting the agents by distance to the centroid of the agent centers. We will add these details on our evaluation setting to the paper.

---

> > ### Comment · Reviewer_KYot · 2023-11-20
> > **Thank you for the clarifications**
> >
> > Thank you for the clear and detailed response to all points raised in the review. I think the new results and clarifications address all of my initial concerns, and I do not have any follow-up questions for now.
> >
> > I look forward to the promised updated version and will decide on my final score once this is available.

---

### Author Response · Authors · 2023-11-16
**Results on the Waymo "Sim Agents" Benchmark**

We thank all reviewers for their reviews. We are incorporating their feedback into our paper and will post direct responses to each of the specific comments and questions from the reviewers soon.

The main critique shared by all three reviewers was that although we compared against baselines based on adjustments to our proposed method, we did not include a direct comparison to prior work in traffic modeling. Towards this end, we have made a submission to the Waymo “sim agents” leaderboard, a benchmark for closed-loop traffic models. Our model is currently in 2nd place, just behind MVTE which is a 3-model ensemble [1]. We note that we are well ahead of the single-model variant of MVTE which is “MVTA”, as well as all other single-model submissions.

|             Method Name                  | Realism Meta metric (test set) | |
|-------------------------------|-----|-----|
| Random (Gaussian) | 16.30 |
| Constant Velocity | 25.76 |
| SBTA-ADIA | 39.40 |
| Wayformer (Identical Samples) | 42.50 |
| CAD | 43.21 |
| MTR+++ | 46.97 |
| Wayformer (Diverse Samples) | 47.20 |
| Joint Multipath++ | 48.88 |
| MTR_E | 49.11 |
| G-net| 49.36 |
| SceneDM | 50.00 |
| SceneDMF | 50.60 |
| MVTA| 50.91 |
| **Trajeglish** | **51.54** |
| MVTE* | 51.70 |
"*" indicates ensembled version of MVTA

Trajeglish surpasses the performance of models built on several well-established previous motion prediction models; we outperform Wayformer by 9.2%, MultiPath++ by 5.6%, and MTR (which the MVTA model is based on) by 1.3%. Our model is the first submission to the leaderboard that uses discrete sequence modeling in contrast to diffusion or mixture-of-x modeling.

We additionally report the component-wise comparison of the leaderboard metrics between Trajeglish and MVTA/MVTE. We highlight that Trajeglish surpasses the previous state of the art for realistic interaction between agents by 4.4%.

|             Method Name                  | Kinematic Metrics ($\uparrow$) | Interactive Metrics ($\uparrow$) | Map-based metrics ($\uparrow$) | minADE ($\downarrow$) |
|-------------------------------|-----|-----|-----|-----|
| MVTA | 41.75 | 51.86 | 63.74 | 1.8698 m |
| Trajeglish | 39.15 | **55.21** | **65.09** | 1.6825 m |
| MVTE* | **42.02** | 52.89 | 64.86 | **1.6770** m |
"*" indicates ensembled version of MVTA

In order to make a submission to this benchmark, we made a few very minor adjustments to the hyperparameters of the model that we have used for most of the analysis and ablation in our paper. We are editing our paper to document these adjustments. At inference time, we sample the 32 scenarios required of submissions to the benchmark in the same way that we sample scenarios for the rest of the paper, e.g. by sampling an action for each agent from the categorical distribution output by our model using a fixed temperature and nucleus that were optimized for the leaderboard metric using the validation set. We are excited to present these results and look forward to discussing the them as well as the remaining questions and comments from the reviewers during the remainder of the discussion period.

[1] “Multiverse Transformer: 1st Place Solution for Waymo Open Sim Agents Challenge 2023”, Yu et al, 2023.

---

> ### Author Response · Authors · 2023-11-23
> **Updated paper**
>
> We thank the reviewers for their helpful feedback. We have updated our paper to account for the edits suggested by reviewers. We highlight the changes below:
> - We added videos for 20 second rollouts sampled from our model to the supplementary material (long_rollouts.mp4). We encourage all reviewers to view these. We also include a selection of rollouts from the nuScenes dataset (nuscenes_rollouts.mp4). (R3)
> - We added a visualization of the embeddings learned by the transformer for each of the tokens to showcase that the model learns the geometric similarity between tokens despite the use of a discretized representation (Figure 14) (R3)
> - We added a preliminary study of how our model scales as a function of parameter count and dataset size (Figure 15). It appears our model is heavily data-constrained - gains are projected to be much more significant from doubling the data size compared to doubling the model size. (R1)
> - We moved related work to main body of the paper, and added MotionLM to the related work (R1+R2)
> - We added a description of how we sampled our submission to the sim agents benchmark from our model in Appendix A.4 and added Table 1 which documents our results on the WOMD sim agents benchmark, as we previously summarized for reviewers in our note above (R1+R2+R3).
> - We added a plot of the full distribution of per-timestep discretization errors for each of the 4 tokenizers we compared against (Figure 13) (R3)

---

### Meta-Review · Area_Chair_GB9D · 2024-01-07

**Metareview:**

This submission suggested a new approach for driving trajectory prediction based on discrete tokenization and transformer. With a simple set of tokenized vocabularies of short sequences and merging it with a transformer (inspired from language), the results were shown to be strong. All reviewers agreed that the suggested method is novel, and the AC also agrees with this point and also thinks this may potentially help in other domains involving sequences. There were a few concerns regarding results and missing baselines, but the authors addressed them all well in the rebuttal. Thus, with this competitive performance and also a novel approach, the submission is suggested to be accepted.

**Justification For Why Not Higher Score:**

- While the tokenization+transformer is new in this domain and the performance is strong, the gain is limited and it is unclear how well it will scale to other domains.

**Justification For Why Not Lower Score:**

n/a

---

### Decision · Program_Chairs · 2024-01-16

Accept (poster)